# Tumor Milieu Controlled by RB Tumor Suppressor

**DOI:** 10.3390/ijms21072450

**Published:** 2020-04-01

**Authors:** Shunsuke Kitajima, Fengkai Li, Chiaki Takahashi

**Affiliations:** 1Department of Cell Biology, Cancer Institute, Japanese Foundation for Cancer Research, Tokyo 135-8550, Japan; 2Department of Medical Oncology, Dana-Farber Cancer Institute, Boston, MA 02215, USA; 3Division of Oncology and Molecular Biology, Cancer Research Institute, Kanazawa University, Ishikawa 920-1192, Japan; cyperus@stu.kanazawa-u.ac.jp (F.L.); chtakaha@staff.kanazawa-u.ac.jp (C.T.)

**Keywords:** retinoblastoma, tumor microenvironment, reactive oxygen species, CCL2

## Abstract

The *RB* gene is one of the most frequently mutated genes in human cancers. Canonically, RB exerts its tumor suppressive activity through the regulation of the G1/S transition during cell cycle progression by modulating the activity of E2F transcription factors. However, aberration of the *RB* gene is most commonly detected in tumors when they gain more aggressive phenotypes, including metastatic activity or drug resistance, rather than accelerated proliferation. This implicates RB controls’ malignant progression to a considerable extent in a cell cycle-independent manner. In this review, we highlight the multifaceted functions of the RB protein in controlling tumor lineage plasticity, metabolism, and the tumor microenvironment (TME), with a focus on the mechanism whereby RB controls the TME. In brief, RB inactivation in several types of cancer cells enhances production of pro-inflammatory cytokines, including CCL2, through upregulation of mitochondrial reactive oxygen species (ROS) production. These factors not only accelerate the growth of cancer cells in a cell-autonomous manner, but also stimulate non-malignant cells in the TME to generate a pro-tumorigenic niche in a non-cell-autonomous manner. Here, we discuss the biological and pathological significance of the non-cell-autonomous functions of RB and attempt to predict their potential clinical relevance to cancer immunotherapy.

## 1. Introduction

The retinoblastoma (RB) tumor suppressor protein plays a pivotal role in the control of cell cycle, terminal differentiation, and various other biological events. RB is genetically or functionally inactivated in many human cancers, including retinoblastoma, small cell lung cancer (SCLC), prostate cancer, and breast cancer [1,2,3,4]. The canonical pathway whereby RB exerts its tumor suppressive activity entails the formation of a transcriptional repression complex with E2F transcription factors and various chromatin modifiers, such as histone deacetylases (HDACs). This complex orchestrates the G1/S transition during cell cycle progression primarily by controlling E2F target genes [1,2,3,4]. Most mitogenic signals commonly merge on the transcriptional upregulation of D-type cyclins and then stimulate cyclin-dependent kinases, including CDK4/6. D-type cyclin-CDK4/6 complexes have been proposed to promote mono-phosphorylation on RB, which allow it to exert early G1 functions by starting the release of E2Fs [1,5,6,7]. E2F target genes, including cyclin E and A, in cooperation with CDK2 or CDK1, are responsible for full phosphorylation of RB at 13 remaining sites. This allows cells to enter the S and M phases [1,5,6,7]. Since uncontrolled cell proliferation is a hallmark of cancer cells, it has been postulated that genes inhibiting RB function, including *CCND*1 and *CDK*4, act as oncogenes. Conversely, genes activating RB function, including cyclin-dependent kinase inhibitors (e.g., *CDKN1A*, *CDKN1B*, and *CDKN2A*), are well-known to act as tumor suppressor genes. The fact that genetic and/or epigenetic aberrations of the components in the RB pathway tend to be mutually exclusive in the patients [8,9] might implicate a linearity of the RB pathway (Figure 1).

Over several decades, many researchers have attempted to pharmacologically target the aberrant status of the RB pathway in cancer cells. Most compounds developed early on did not achieve clinical benefits due to the lack of selectivity and potency. However, current highly-specific and potent CDK4/6 inhibitors, such as palbociclib (PD0332991), ribociclib (LEE011), and abemaciclib (LY2835219), have shown promising efficacy in patients, especially with RB-intact estrogen receptor (ER)-positive and HER2-negative breast cancer [10,11]. This led US Food and Drug Administration (FDA) to endow accelerated approval to these drugs as breakthrough therapies in 2015. Guidelines recommend using these in combination with an aromatase inhibitor or estrogen antagonist. The therapeutic efficacy of these inhibitors depends mostly on the long-term maintenance of the unphosphorylated status of the RB protein. Therefore, loss-of-function *RB* mutations would result in a gain of resistance to the treatment with CDK4/6 inhibitors. Even in the presence of intact RB, many events (e.g., FAT1 loss, Cyclin E1 or CDK6 overexpression, PI3KCA mutation) were reported to cause resistance to these compounds in breast cancers [12]. The emergence of novel CDK4/6 inhibitors could be praised as one of the triumphs achieved by RB research. However, understanding the methods for maximizing the utility of these compounds and resolving the resistance to them remains necessary.

In addition to the canonical RB pathway driven by its interaction with E2Fs and HDACs, the RB protein also has functions independent of E2Fs, acting as transcriptional activators. For instance, chromatin immunoprecipitation and sequencing (ChIP-seq) revealed that the RB protein can also bind to intronic and intergenic regions as well as in promoters with the E2Fs-binding sites [13,14,15]. One of the most well-established, non-canonical functions of RB is to maintain genome stability during DNA replication and mitosis [16]. For example, the RB-E2F complex recruits condensin II to secure chromosomal condensation and subsequent DNA segregation [17]. RB inactivation, therefore, induces aneuploidy. In fact, according to a comprehensive genomic analysis of the human tumor genome, genomic instability tends to be higher in tumors with mutations in the RB pathway [17,18]. Recent studies have shown that inhibition of kinases related to cell cycle check points, including checkpoint kinase 1 (CHK1), polo-like kinase 1 (PLK1), or aurora kinase A or B, exhibits synthetic lethality in combination with RB deficiency in triple-negative breast cancer (TNBC) or SCLC [19,20]. This indicates that genomic instability in RB-deficient cancer cells may be a pharmacologically vulnerable target. In addition to its role in genome stability, it is becoming increasingly clear that RB possesses multifaceted functions in controlling cell death, differentiation, metabolism, stemness, and innate immune signaling [3,16,21,22,23,24,25,26]. To date, more than 300 proteins have been reported to bind with the RB protein. The variability in these binding partners could explain the multifunctional aspect of the RB protein. In this article, among such a variety of RB functions, we focused on those known to regulate lineage plasticity, cancer metabolism, and inflammatory signaling. We extended the discussion toward understanding how these functions allow RB to orchestrate the tumor microenvironment (TME) through the regulation of inflammatory signaling.

## 2. Beyond Cell Cycle Regulation

### 2.1. Increased Lineage Plasticity Induced by RB Inactivation

Although RB is primarily implicated in the regulation of the cell cycle, *RB* mutation is frequently observed in late-stage cancer or at metastatic sites in which uncontrolled cell proliferation is likely to be established prior to RB inactivation [16]. It has been reported that RB promotes differentiation that is independent of the cell cycle regulation and RB-inactivated cells, therefore, exhibit defective terminal differentiation [1,2,3]. Interestingly, aberration of the *RB* gene often correlates with appearance of phenotypes associated with dedifferentiation or transdifferentiation in lung cancer, prostate cancer, and breast cancer [27,28,29,30,31]. The lineage plasticity induced by RB inactivation would promote the resistance to therapies by epidermal growth factor receptor (EGFR) inhibitors, estrogen and androgen receptor antagonists, and androgen deprivation (castration) because these treatments generally target cell lineage-specific characteristics of tumors [28,29,32,33,34]. Several groups have reported that simultaneous inactivation of multiple RB family members (e.g., RB, RB2/p130, and RB3/p107) induces not only cell cycle re-entry but also increases lineage plasticity in post-mitotic cells. For example, mouse embryonic fibroblasts (MEFs), in which all RB family proteins are inactivated, show a resistance to G1 cell cycle arrest and acquire characteristics similar to those of stem cells, as depicted by elevated sphere-forming activity and expression of pluripotent genes [35]. RB depletion in an *ARF*-null genetic background induces cell cycle re-entry and dedifferentiation in post-mitotic muscle cells [36]. Moreover, the RB-E2F1 complex suppresses the expression of pluripotent factors, such as *SOX2* and *POU5F1*, by directly binding to their regulatory regions [14]. Consistent with these reports, RB inactivation increases the generation efficacy of inducible pluripotent stem (iPS) cells from human fibroblasts [14]. In prostate cancer cells, RB inactivation increases cellular plasticity, which increases resistance to anti-androgen therapy and promotes metastasis via induction of SOX2, especially in a *TP53*-deficient background [29,30]. Another paper revealed that RB depletion, together with p53 inactivation, induces transdifferentiation from adenocarcinoma to neuroendocrine-type cancers in prostate and lung cancer cells [37]. Mice heterozygous for *Rb* generate thyroid medullary cancer (MTC) from calcitonin-producing neuroendocrine cells as a consequence of biallelic loss of *Rb*. We reported that thyroid medullary cancer derived from calcitonin-producing cells as a consequence of biallelic loss of *Rb*; these cells exhibit dedifferentiated characteristics, depicted by a lower expression of calcitonin in a *Trp53*-null but not in genetic backgrounds lacking other p53 pathway genes, such as *Arf* and *Cdkn1a* [38]. Many papers have described that loss of p53 function synergizes with RB inactivation towards upregulation of cancer lineage plasticity. In prostate cancer cells, it has been demonstrated that RB and p53 cooperate to suppress the transcription of epigenetic reprogramming factors, such as SOX2 and EZH2 [29,30]. The p53 pathway is activated following RB inactivation depending on E2F-1 and their target ARF, which may provide another mechanism underlying the synergism between RB and p53 inactivation [39]. p53 appears to strongly counterbalance the effects of RB loss on the cell cycle, survival, and presumably many unidentified cellular behaviors; thus, p53 inactivation seems to promote a malignant phenotype induced by RB inactivation. Indeed, in some fraction of cancers, unique phenotypes are observed only with double inactivation of both RB and p53 [27,40,41,42,43].

### 2.2. Metabolic Reprogramming Induced by RB Inactivation

Since cell cycle progression requires the production of large amounts of energy and biomass for cell division, it is not surprising that RB regulates metabolic pathway involving glycolysis, the TCA cycle, and oxidative phosphorylation (OXPHOS) as well as cell cycle progression. In general, rapidly proliferating cells, including cancer cells, require a shift to anabolic metabolism in order to supply nucleotides, amino acids, and fatty acids for the synthesis of DNA/RNA, protein, and lipid membranes to produce daughter cells [44,45]. Of note, many oncogene and tumor suppressor proteins, including RAS, PTEN, and p53, directly regulate multiple metabolic pathways [46,47,48] and glutaminolysis is especially a well-studied pathway regulated by RB-E2Fs. For example, RB depletion increased glutamine consumption through E2F-dependent upregulation of a glutamine transporter encoded by the *ASCT2*/*SLC15A* gene [49]. In addition, Rb depletion in the fly promotes glutamine metabolism [50]. Moreover, the expression of pyruvate dehydrogenase kinases (PDKs), which inhibit the conversion of pyruvate to acetyl-CoA, is negatively regulated by E2F-1 in cooperation with histone demethylase KDM4A [51]. Hence, RB depletion causes cells to be more reliant on glutaminolysis rather than glycolysis for supplementation of carbon sources to the TCA cycle and OXPHOS [51]. The biological significance of upregulated glutaminolysis in RB-depleted cells is under debate; cells may have an increased necessity to synthesize glutathione, a representative intracellular antioxidant, from glutamine to quench reactive oxygen species (ROS) that are excessively produced as a consequence of accelerated cell proliferation induced by the RB depletion.

## 3. Enhanced Cytokines Secretion by Oxidative Stress Induced by RB

### 3.1. Increased Cellular Stress Induced by RB Inactivation Promotes Cytokine and Chemokine Secretion

In general, accumulation of cellular stresses caused by rapid proliferation and high metabolic demand stimulates production of a variety of growth factors, cytokines, and chemokines, which act on cell homeostasis. The variance among these factors depends on the cellular context. The accumulation of multiple kinds of cellular stress (e.g., DNA damage, replicative stress, oxidative stress, ER stress), likely to be induced by aberration of the RB pathway, might affect the production of growth factors, cytokines, and chemokines from cancer cells. We demonstrated that RB inactivation forces the mitochondria to produce higher levels of ROS, leading to the accumulation of oxidative stress and subsequent increased secretion of cytokines [52]. Cancer cells commonly depend on glycolysis rather than OXPHOS for energy production, which is known as the Warburg effect [53]. Warburg speculated that cancer cells might depend less on mitochondrial activity for survival. However, mitochondrial metabolism is indispensable for the growth of cancer cells [54]. OXPHOS is typically upregulated in cancer cells compared to that in normal cells, which is considered to enable cells to generate enough ATP to drive a variety of ATP-demanding anabolic pathways. Treatment with inhibitors targeting OXPHOS therefore suppresses the growth of cancer cells [55]. Higher ATP production contributes not only to rapid cell proliferation but also to enhanced cell motility and dissemination [25]. Of note, RB/E2Fs directly regulate mitochondrial protein translation and OXPHOS. E2F-1 directly binds to the promoter and induces the expression of mitochondrial protein-coding genes, such as *MRPL37* and *TOMM4* [56]. RB depletion in breast cancer cells increased the expression of mitochondrial proteins and stimulated cytokine secretion depending on the elevated mitochondrial ROS production [52]. RB depletion in breast cancer cells increased production of multiple cytokines and chemokines, including IL-6 and CCL2. The involvement of mitochondrial ROS in this phenomenon was demonstrated by a rescue experiment using antioxidants, including N-acetyl-L-cysteine (NAC) and mitoubiquinone (MitoQ) [52].

It has been reported that RB is required for the induction of autophagy that occurs under a hypoxic condition [57,58]. RB-inactivated cells therefore show the accumulation of damaged mitochondrial due to the failure in proper scavenging of dysfunctional mitochondria by mitophagy, which may lead to increased ROS production [59]. In general, accumulation of damaged or aged mitochondria is supposed to cause higher mitochondrial ROS production due to an insufficiency to quench the superoxide produced by OXPHOS [60]. In fact, comprehensive proteomic analysis of RB-inactivated cells identified functional abnormalities in the mitochondria [61]. The expression of pro-inflammatory cytokines is also directly regulated by the RB-E2F complexes as well. For example, activation of E2Fs following RB inactivation directly induces PTGS2, prostaglandin-endoperoxide synthase 2, in basal-like breast cancer cells [62]. Furthermore, RB depletion induces IL-6 secretion in *Cdkn1a*-deficient epidermis [63]. E2F binding sites were found to reside in the regulatory region of genes of a series of cytokines and their receptors; the RB-E2F complex suppresses the transactivation of genes for cytokine and chemokine, including CXCL1, CXCL2 and IL-8 [15]. To date, several reports have revealed that RB inactivation is associated with enhanced cytokine expression in several types of cancer cells. This may explain why RB inactivation enhances hormone-independent growth and metastasis of tumor cells.

### 3.2. RB Is Involved in the Mechanism to Keep Redox Balance at a Nontoxic Level

It is well-established that pro-inflammatory cytokines, such as IL-6 and IL-8, play crucial roles in the generation and maintenance of the stem cell-like fraction in tumor cells through stimulating NF-κB and STAT3 signaling, which promotes therapy resistance, tumor recurrence, and metastasis [59,64]. For example, administration of IL-6 or IL-8 is sufficient to induce mammosphere-forming activity in luminal-type breast cancer cells [65,66]. Similarly, several studies demonstrated that activation of the IL-6-STAT3 pathway induces expansion of the stem cell-like population and acquisition of drug resistance, including hormone-independent growth in breast cancer cells [67,68,69,70,71]. The IL-6-STAT3 pathway is reported to contribute to acquisition of drug resistance in lung and liver cancer cells as well [72,73,74].

A number of cytokines that are induced by RB inactivation seem to share an antioxidative function, which is required to overcome the excessive ROS accumulation. The treatment of RB-deficient breast cancer cells with an antibody that is capable of neutralizing IL-6 activity accelerated the accumulation of mitochondrial superoxide and subsequent cell death [52]. In contrast, the treatment of breast cancer cells with recombinant IL-6 diminished mitochondrial superoxide production, which was associated with increased mammosphere-forming activity and hormone-independent growth [52]. These findings indicate that RB controls mitochondrial superoxide production through bivalent pathways: one involves mitochondrial ROS and the other involves IL-6, which counterbalances the former to keep the redox balance at a nontoxic level. Consistent with antioxidative function of IL-6, the promoter region of the *IL-6* gene contains an antioxidant responsive element (ARE), which is a binding sequence for NRF2, a master regulator of the transcriptional response to oxidative stress [75]. On the other hand, NRF2 can interfere with the induction of pro-inflammatory cytokines from macrophages [76]. The role of NRF2 in RB-mediated control of cytokine secretion and the involvement of oxidative stress in it requires further debate.

Investigation of the mechanism whereby RB controls IL-6 production indicated that RB inactivation stimulates fatty acid oxidation (FAO) and increases ROS production from mitochondria, which then activates the Jun kinase (JNK) pathway. Mechanistically, RB inactivation increased AMP-activated protein kinase (AMPK) phosphorylation and subsequent acetyl-CoA carboxylase (ACC) phosphorylation, resulting in the downregulation of malonyl-CoA synthesis. Decrease in malonyl-CoA allows CPT1 to transport long-chain fatty acids into the mitochondria for FAO. This pathway is demonstrated by the antagonistic effect of JNK inhibitors, FAO inhibitors, and antioxidants on IL-6 upregulation induced by RB inactivation [52]. In addition, downregulation of microRNA-140 (mir-140) following Rb inactivation in mouse sarcoma cells resulted in the upregulated expression of multiple pro-inflammatory cytokines, including Il-6, Vegfα, and other growth factors that have a mir-140 binding sequence in their 3’ UTR. Although the molecular mechanism that underlies the regulation of mir-140 by RB is still unclear, we speculate that RB functions to destabilize the mRNA of these factors through micro RNAs [77]. Collectively, we proposed a novel role for RB in controlling the production of pro-inflammatory cytokines: this pathway controls metabolic reprogramming leading to FAO alteration, mitochondrial ROS, JNK pathway, and antioxidative pathways to maintain cellular redox balance at a nontoxic level in a cell-autonomous manner.

## 4. RB Impacts the Tumor Microenvironment via Chemokine

In tumor tissues, elevated cytokine secretion following RB inactivation may stimulate not only cancer cells themselves but also the surrounding non-malignant cells, called TME, which include immune cells, fibroblasts, and vascular networks. CCL2 is a well-characterized chemokine that stimulates the infiltration of its receptor CCR2-positive cells, including monocytes and/or macrophages, into the TME. The recruitment of these immune cells further stimulates angiogenesis to nourish cancer cells mostly in a vascular endothelial growth factor (VEGF)-dependent manner [78]. Recent studies indicated that CCL2 plays a critical role in triggering the infiltration of immunosuppressive cells, such as myeloid-derived suppressor cells (MDSCs) and regulatory T-cells (T-regs), causing a shift to pro-tumorigenic TME [79,80]. Accordingly, many groups are currently attempting to control malignancies by the pharmacological blockade of CCL2-CCR2.

We once established a ‘soft tissue sarcoma system’, derived from a well-differentiated leiomyosarcoma developed in a *Trp53*-null mouse, in which additional Rb inactivation induces a phenotypic change to undifferentiated type. We identified significant upregulation of CCL2 and a pro-angiogenic cytokine Vegfα following Rb depletion in this system [81]. Interestingly, mouse soft tissue sarcoma cells did not express CCR2. Consistent with higher secretion of CCL2 and Vegfα, upon orthotopic tumor engraftment, the TME of Rb-depleted cells showed higher angiogenesis and infiltration of CCR2-positive tumor-associated macrophages (TAMs) and MDSCs. These phenotypes were significantly suppressed when Rb-depleted cells were xenografted to *Ccr2*-null mice. Interestingly, infiltration of T-regs into the TME was upregulated as well following RB depletion in a CCL2-CCR2-independent manner. This indicated a possibility that RB controls chemokines other than CCL2. Taken together, these observations indicate that RB exerts non-cell autonomous functions through the regulation of cytokine secretion (Figure 2).

CCL2 is upregulated following RB inactivation in human breast cancer cells, not only in mouse sarcoma cells. We observed elevated CCL2 expression in the mammary glands of these mice regardless of the *Ccr2* genotype. We detected hyperplastic growth and prominent infiltration of macrophages in the mammary gland of *Ccr2*^+/+^; *MMTV*-*Cre*; *Rb*^flox/flox^ mice. However, these phenotypes were less significant in *Ccr2*^-/-^; *MMTV*-*Cre*; *Rb*^flox/flox^ mice, suggesting the importance of CCL2-CCR2 in carcinogenesis caused by RB deficiency. In addition to CCL2, we identified several other cytokines and chemokines as possible targets of RB, which may contribute to the foundation of pro-tumorigenic TME as well. As is in case of IL-6, the mechanism whereby RB controls CCL2 seems to entail mitochondrial ROS production and the JNK pathway (Figure 3). Further studies are required to uncover the molecular mechanisms underlying the regulation of cytokines and chemokines by RB.

## 5. Therapeutic Strategy Targeting Innate Immune Signaling in RB-Inactive Cancer

Cancer immunotherapy with immune checkpoint blockade (ICB) using anti-PD-1/PD-L1 or anti-CTLA4 antibodies enhances antitumor activity of the immune components of the TME, including cytotoxic CD8 positive T cells and natural killer cells [82]. ICB exhibits significant therapeutic efficacy for many types of cancer patients, including non-small cell lung cancer (NSCLC), melanoma, and renal cell carcinoma. However, the benefits of ICB are limited for the majority of patients and predictive markers of sensitivity to ICB are still under investigation [83,84,85].

Given that RB inactivation in cancer cells promotes the infiltration of immunosuppressive cells, such as MDSCs and TAMs, into the TME, at least in part through the CCL2-CCR2. RB inactivation due to genetic mutation or transcriptional suppression via DNA hypermethylation in the *RB* gene promoter could be one of potential markers of a so-called ‘cold tumor’, which exhibits poor immunogenicity leading to the resistance to ICB. If so, treatment inhibiting CCL2-CCR2 using agents, such as neutralizing antibody to CCL2, might cooperate with ICB. Since cancers prevalently carry inactivation of the RB pathway, a significant fraction of cancers might respond to such therapy. Interestingly, it has been recently reported that *RB* mutation is correlated with the resistance to anti-PD-1 therapy in advanced NSCLC patients using nivolumab and pembrolizumab [86]. Meanwhile, several papers reported that RB-inactive cells show less responsiveness to the stimuli activating the anti-viral response driven by viral double-stranded RNA or interferon (IFN) gamma [87,88]. Inversely, the treatment with CDK4/6 inhibitor, which reactivates RB through keeping it in an unphosphorylated status, promotes cytokines secretion related to the anti-viral response and reduces the infiltration of immunosuppressive immune cells, such as MDSCs and T-regs [89,90,91]. Since the anti-viral response has recently emerged as a key pathway of antitumor immunity [92,93,94], downregulation of the anti-viral response by RB inactivation might contribute to the resistance to ICB via immune evasion as well, although the molecular mechanisms of how RB regulates anti-viral pathways are not yet completely elucidated.

In general, cellular senescence is strongly dependent on the RB pathway. Senescent cells show a marked increase in the secretion of multiple cytokines, so-called the senescence-associated secretory phenotype (SASP) [95]. Interestingly, cytokines induced during SASP largely overlap with those that are upregulated following the treatment of CDK4/6 inhibitors. However, it is not yet certain whether they constitute a part of the senescence program [96]. A recent study has shown that the RB protein, phosphorylated by CDK4/6-cyclin D complex, specifically suppresses NF-κB p65 activity through direct binding and thereby inhibits the expression of NF-κB targets, including PD-L1 [97], which suggests another mechanism that promotes immune evasion in *RB*-deficient cancer cells.

It has recently become clear that the accumulation of DNA damage in tumors resulting from disruption of the DNA replication/repair pathways, DNA-damaging chemotherapy, or radiation therapy promotes an antitumor immune response [98]. In general, aberrant cytoplasmic DNA accumulation or micronuclei formation following DNA damaging events are detected by cyclic GMP-AMP synthase (cGAS), which produces the second messenger cyclic GMP-AMP (cGAMP) that directly activates its downstream stimulator of interferon genes (STING) [99]. Activation of STING subsequently induces upregulation of its downstream cytokines, including type I interferon and CXCL10, which promotes T-cell-mediated therapeutic antitumor immunity via enhancing neoantigen presentation and T-cell recruitment into the TME [100]. Not only DNA damage accumulation but also excessive ER stress activate the STING pathway [101]. Considering these mechanisms, it is not surprising that the increased genomic instability and metabolic stress that accumulates in RB-inactive cancer cells potentiate the efficacy of immunotherapy in certain contexts. Because of such complexity in the interaction between the RB pathway and innate immune signaling, continued research is needed to elucidate how RB impacts tumor immunogenicity through cell-autonomous and non-cell-autonomous functions.

## 6. Role of Other Tumor Suppressor Proteins in the TME

Based on the extensive analysis of datasets obtained from clinical studies, researchers have identified several markers that are highly predictive of favorable T-cell response. These include higher PD-L1 expression, the degree of tumor mutation burden (TMB), and the number of tumor-infiltrating lymphocytes (TILs) residing in the TME [83,84,85]. In addition, somatic mutations in cancer cells, including loss-of-function mutations in *JAK1/2*, *APLNR*, *PTPN2*, and *PBRM1*, significantly correlate with the efficacy of cancer immunotherapies [102,103,104,105,106,107,108].

As mentioned above, it is likely that *RB* mutation in cancer cells might predict the sensitivity to ICB for certain types of cancer. We here refer to an example that tumor suppressors other than RB can also impact the TME through the regulation of cytokine and chemokine secretion. Liver kinase B1 (LKB1) is a well-known tumor suppressor protein, which is mutated or deleted in Peutz–Jeghers syndrome (PJS) and in a variety of cancer cells, such as NSCLC, breast cancer, and pancreatic cancer [109]. LKB1 is a serine/threonine kinase, which phosphorylates 13 members of the AMP-activated protein kinase (AMPK) family, thereby affecting multiple cellular functions that control cellular metabolism, cell cycle progression, apoptosis, and cell polarity, especially under nutrient-deprived conditions [109]. In general, activation of the LKB1-AMPK pathway contributes to the maintenance of energy homeostasis through multiple mechanisms, such as inhibition of anabolic metabolism, to save ATP consumption and to induce autophagy-mediated degradation of organelles for recycling nutrients [110]. Furthermore, the LKB1-AMPK pathway maintains energy homeostasis by promoting the clearance of damaged mitochondria via selective autophagy, so called mitophagy [111]. Accordingly, LKB1-inactive cancer cells accumulate significant levels of metabolic stress and oxidative stress due to the accumulation of damaged mitochondria, resulting in vulnerability to excessive metabolic stress induced by treatment with mitochondrial inhibitors, such as phenformin [112]. Indeed, addition of the *LKB1* mutation to *KRAS* mutant NSCLC cells is significantly correlated with the aberrant accumulation of cytoplasmic mitochondrial DNA and higher mitochondrial ROS production [113]. Similar to RB-depleted cells that exhibit higher pro-inflammatory cytokines production due to increased mitochondrial ROS, *KRAS*;*LKB1*-mutated NSCLC cells highly produce cytokines, such as IL-6 and CCL5 [114,115]. Higher secretion of these pro-inflammatory cytokines not only promotes cell growth and cell survival in a cell-autonomous manner but also contributes to the formation of immunosuppressive TME in a non-cell-autonomous manner [114,115]. On the other hand, aberrant accumulation of cytoplasmic DNA derived from mitochondria in *KRAS*;*LKB1*-mutated NSCLC cells stimulates the cGAS-STING pathway, as discussed above. *KRAS*;*LKB1*-mutated NSCLC cells seem to need to attenuate the STING pathway during tumor evolution to protect the cells from cell death caused by intrinsic signals or enhance the recruitment of cytotoxic T cells to remove pathogenic cells [116]. Indeed, expression of STING and its downstream cytokines are significantly lower in *KRAS*;*LKB1*-mutated NSCLC patients than in *LKB1* intact ones [113]. Consistent with these findings, *KRAS*;*LKB1*-mutated NSCLC patients exhibited lower infiltration of cytotoxic T cells into the TME and resistance to anti-PD-1 therapy [117] possibly because the secretion of type I interferon and CXCL10 downstream of STING plays a critical role in the anti-tumorigenic response following ICB.

## 7. Conclusions

Cell cycle dysregulation is one of the most important features caused by RB inactivation. In addition, RB exerts multifaceted functions and regulates many other biological events, including genome integrity, lineage plasticity, cellular metabolism, and innate immune signaling. The malignant phenotypes induced by RB inactivation can be context-dependent and the molecular mechanisms underlying this aspect are not fully understood. It is not surprising that, even after more than 30 years of investigation, RB still defies our full understanding of its function. As mentioned above, RB may have more than 300 binding partners. RB may select its binding partners depending on cell type and cellular context. In this review, we focused on the interaction between the RB pathway and innate immune signaling. To date, many studies have revealed that the status of RB affects cytokines/chemokines secretion directly or indirectly in an E2F-dependent or -independent manner. This suggests a novel RB function in controlling intracellular signaling, which may be exerted through non-cell-autonomous regulation of innate immune signaling. Recently, we have reported a novel function of RB in controlling the CCL2-CCR2 function, which will enable us to predict the nature of the TME depending on the RB status in tumor cells. This finding may provoke a novel debate on the utility of blockers of the CCL2-CCR2 pathway with an aim toward enhancing the sensitivity to ICB.

## Figures and Tables

**Figure 1 ijms-21-02450-f001:**
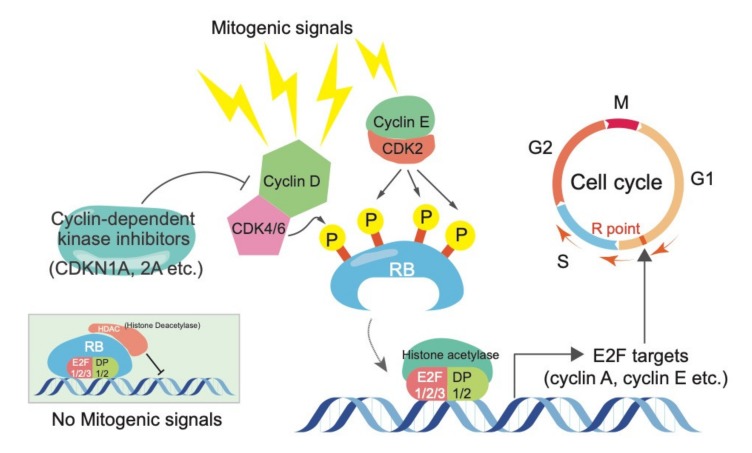
The canonical RB function in controlling cell cycle progression. Cyclin-CDK complexes regulate RB activity via phosphorylation, while cyclin-dependent kinase inhibitors suppress the activity of Cyclin-CDK complexes. Phosphorylated RB releases E2Fs, which stimulate cells to enter to the S and M phases via induction of E2F targets, such as cyclin A and E.

**Figure 2 ijms-21-02450-f002:**
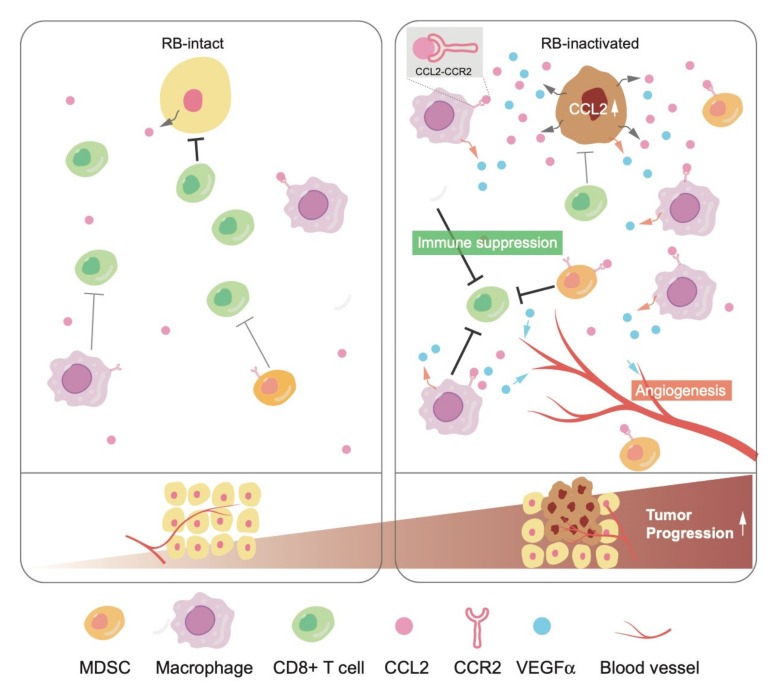
RB impacts the TME via cytokine and chemokine secretion. RB inactivation upregulates CCL2 and VEGFα secretion, which recruits immune suppressive cells, such as MDSCs and tumor-associated macrophages and promotes angiogenesis.

**Figure 3 ijms-21-02450-f003:**
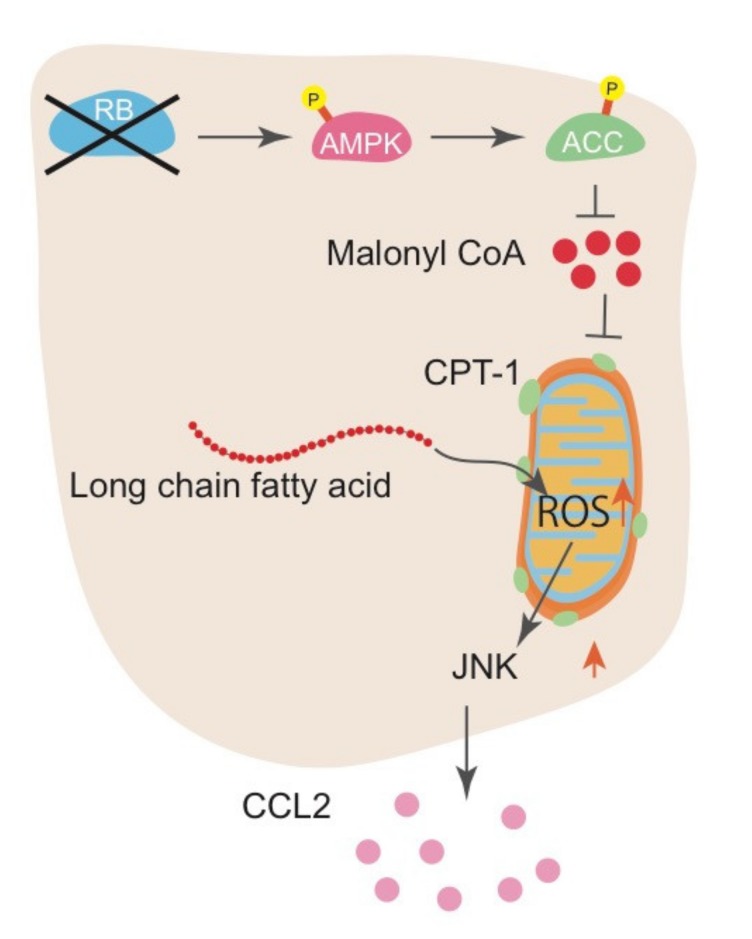
The mechanism of CCL2 upregulation following RB inactivation. RB inactivation increases AMPK phosphorylation and subsequent ACC phosphorylation, resulting in downregulation of malonyl-CoA synthesis. A decrease in malonyl-CoA allows CPT1 to transport long-chain fatty acids into mitochondria for FAO, promoting mitochondrial ROS production.

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
