# Peer review of "Tumor Milieu Controlled by RB Tumor Suppressor"

_ijms, 2020, doi:10.3390/ijms21072450_

Round 1

Reviewer 1 Report

The authors in the current manuscript entitled: "Tumor milieu controlled by Rb tumor suppresor" report the multifaceted role of Rb1 in controlling cell plasticity, metabolism and the tumor microenviroment beyond the cell cycle regulation.

The authors describe as RB defective cells use glutaminolysis to synthesize glutathione to quench excessive production of ROS. Besides high cellular stress induced by RB inactivation promote release of cytokine IL6 which in turn counterbalances the mithoconfrial ROS. RB gene status also controls TME via chemokine CCL2-CCR2 pathway. RB inactivation upregualts CCL2 and VEGF alpha secretion, which results in immunosuppressive effects by recruitment of myeloid-derived suppressors cells (MDSCs), tumor-associated macrophages (TAM) and promotion of angiogenesis. Further studies are required to unveil the molecular mechanisms underlying the regulation of cytokines and chemokines by RB. The authors conclude that all those observations support the hypothesis of using blockers for CCL2-CCR2 in sinergy with immune checkpoints blockade.

Reviewer 2 Report

Contribution of retinoblastoma gene and tumor-microenvironment are very well studied areas in the development therapy of cancer. However, there are very few comprehensive reviews on the interface of both these fields and this review is very timely and would greatly helps researchers in this area. Review is very well written with appropriate references and I did not see any need for modifications and it can be published as it is.